# Center-Line Velocity Change Regime in a Parallel-Flow Square Exhaust Hood

**DOI:** 10.3390/ijerph17124485

**Published:** 2020-06-22

**Authors:** Jianwu Chen, Longzhe Jin, Zhenfang Chen, Bin Yang, Yanqiu Sun, Shulin Zhou

**Affiliations:** 1School of Civil and Environmental Engineering, University of Science and Technology Beijing, Beijing 100083, China; lzjin@ustb.edu.cn; 2Institute of Occupational Health, China Academy of Safety Science and Technology, Beijing 100029, China; czflc7922@126.com (Z.C.); ybustb@163.com (B.Y.); sunyanqiu2002@163.com (Y.S.); zhousl@chinasafety.ac.cn (S.Z.)

**Keywords:** center-line velocity, poison control, parallel-flow, exhaust hood

## Abstract

A parallel-flow exhaust hood is an effective ventilation device to control dust and toxic pollutants and protect the occupational health of workers, whether it is used alone or combined with a uniform air supply hood in a push–pull ventilation system. Some scholars have studied the outside air flow characteristics of the conventional exhaust hood with non-uniform air speed at the hood face, but the law of velocity variation outside the parallel-flow exhaust hood is not clear at present. Therefore, this paper uses the dimensionless method to study the center-line velocity change regime in a parallel-flow square exhaust hood based on simulation and experimental data. The results show that the dimensionless center-line velocity has a good change law with the characteristic length of exhaust hood in a parallel-flow square exhaust hood, which can eliminate the influence of hood face velocity and the hood size on the velocity change regime; and the experimental data is basically consistent with the calculated data, which shows that the regression equation method is reliable.

## 1. Introduction

A parallel-flow square exhaust hood can supply uniform exhaust air, which is different from the general exhaust hood. The uniform flow structure of pipelines has been studied by some scholars [1], but there is little research on the airflow distribution outside a parallel-flow hood. Because of the parallel-flow, it improves the exhaust hood performance during indoor ventilation, and it can also be combined with a uniform supply hood to produce a parallel-flow push–pull ventilation system. Parallel-flow push–pull ventilation systems [2,3] are widely used, because they can solve many technical problems, such as long-distance control of toxicants, and can effectively control dust and toxic pollutants and ensure the occupational health of workers, owing to their special requirements for velocities and energy saving advantages [4,5,6].

Previous studies have analyzed the aerodynamic characteristics, design guidelines [7], performance [8], and field measurements [9] of push–pull ventilation systems, as well as the design of uniform air supply and exhaust [10], balanced ventilation principle and calculation [11], and non-adjustment of static pressure in a supply air duct [12]. A comparison between the main current formulas available in the literature and the experimental results collected by the authors is introduced and discussed for plain and flanged free-standing exhaust slot openings [13]. But the comparison for a parallel-flow square exhaust hood with large hood opening and uniform velocity, which can be used in a parallel-flow push–pull ventilation system, is not discussed. When most of the common formulas presented in a dimensionless form are used in practice for the design and characterization of local exhaust hoods, they allow calibration of only the velocity induced along the opening axis. 

The velocity change regime for a parallel-flow square supply hood [14] and a desktop slot exhaust hood without flange [15] via dimensionless numerical simulation were studied. However, it is not clear whether the center-line velocity of a parallel-flow square exhaust hood has a similar change regime as a parallel-flow supply hood or a desktop slot exhaust hood without flange. A parallel-flow exhaust hood with a large hood opening and uniform velocity is an important part of a parallel-flow push–pull ventilation system, but there is no consensus on the principle of parallel-flow push–pull ventilation system. The center-line velocity change regime for a parallel-flow exhaust hood is helpful to the study of velocity change regime in a parallel-flow push–pull ventilation system, and can further improve the ability of controlling dust and poisonous pollutants for the exhaust hood since its strong anti-interference air flow ability [4,5]. It has a positive effect on reducing the dust and poison in the operating environment and ensuring the occupational health of workers [14,16,17]. Therefore, this study aimed to establish the velocity change regime for a parallel-flow square exhaust hood used in a push–pull ventilation system without a parallel-flow push hood, and the results can provide a theoretical basis for improving the ability of controlling dust and poisonous pollutants for the exhaust hood.

## 2. Materials and Methods

### 2.1. Geometric Models

A 0.5 m (long) by 0.5 m (wide) parallel-flow square exhaust hood was selected as the research object, because the parallel-flow square exhaust hood with this dimension is commonly used as the square exhaust hood minimizes the impact of geometric dimensions. A 25 m (long), 4 m (wide), and 4 m (high) computational domain was established and the exhaust hood was located in the center of the right surface of the computational domain in order to observe the air distribution outside the hood, and the six surfaces of the computational domain were all set to outflow in order to avoid the influence of the space size on the airflow distribution. The mathematical model and the computational domain are shown in Figure 1. 

The exhaust hood was meshed using a TGrid side length of 0.03 m (hybrid grid), and the calculation field was meshed using a TGrid side length of 0.05 m.

### 2.2. Research Conditions

The mathematical model of gas motion was the same as that employed in previous studies [16,17,18]. It was assumed that the air in the flow field is an incompressible Newtonian fluid. The flow was assumed to be a fully developed turbulent flow and the viscosity between fluid molecules was negligible, and only the momentum transfer was considered in this study and heat transfer was ignored. These assumptions coincide with the *k-ε* model, and the standard *k-ε* double-equation model has a reasonable accuracy and has been widely adopted in engineering simulation calculations [18]. Therefore, this study uses a 3D solver and the standard *k-ε* double-equation model.

It is assumed that the hood face velocity is stable and approximately parallel in this study, so this air flow can be considered as a parallel flow as defined by ISO (International Organization for Standardization) 14644.3-2019. The purpose of this study was to research the velocity change regime outside the parallel-flow exhaust hood, so the hood face was set as a velocity inlet in order to ensure a uniform air flow at the exhaust hood face. Velocities of 0.5 m/s, 0.6 m/s, 0.7 m/s, and 0.8 m/s were set as the hood face speeds of the parallel-flow square exhaust hood, as the typical range of exhaust speeds used in push–pull ventilation systems is 0.5–0.8 m/s. The end of the circular duct was set as outflow. The main parameters and boundary conditions of the numerical simulation are shown in Table 1.

The variables used in this study are shown in Figure 2.Figure 2The variables used in this study.
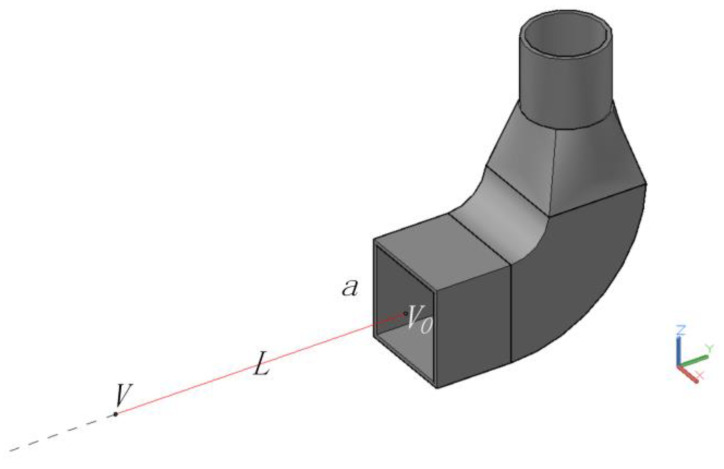
where *a* is the square hood side length; *V*_0_ is the hood face velocity (because the air flow of the parallel-flow exhaust hood studied in this paper is uniform, the velocity at any point of the hood face is same, thus *V*_0_ is also the velocity at the center of the hood face)*; L* is the distance from the hood face; *V* is the velocity at the location *L* form the hood face.

The center-line velocity obtained through the simulation results was analyzed using Excel. Firstly, the changes in the simulation values of the center-line velocity (*V*) at different distances (*L*) from the hood face were plotted. Then, the change in the center-line velocity with distance was investigated using the dimensionless method. Finally, the dimensionless data was fit to a curve. 

## 3. Results

### 3.1. Influence of Hood Face Speed on Velocity Change Regime

The center-line velocity change regime was simulated and determined using Fluent for different hood face velocities (specifically, 0.5 m/s, 0.6 m/s, 0.7 m/s, and 0.8 m/s). The results are shown in Figure 3. 

The center-line velocity decreased rapidly with an increasing distance from the hood face, because the velocity decreased to approximately zero at a distance of 1.5 m from the hood face for all hood face velocities. When the hood face velocity was different, the variation law of the center-line velocity was different, which is consistent with conclusions in previous research [14,15]. Therefore, we used the same dimensionless method as used in previous literature [14,15] to study the velocity change regime and explore whether the similar variation law can be obtained.

### 3.2. Dimensionless Analysis of Velocity Change Characteristics

To eliminate the influence of hood face size and hood face velocity of the exhaust hood, the dimensionless method was used in this study to obtain the center-line velocity characteristics of the parallel-flow square exhaust hood. The dimensionless method is as follows.

The ratio of the distance from the hood face (*L*) to the square hood side length (*a*) is taken as the *x*-axis, and the ratio of the simulation values of velocity (*V*) at different distances from the hood face to the hood central velocity (*V*_0_) is taken as the *y*-axis. The dimensionless data obtained are plotted as shown in Figure 4.

Whether it was a parallel-flow square exhaust hood studied in this paper, a parallel-flow square supply hood studied Chen et al. [14], or a desktop slot exhaust hood without flange studied in Wu et al. [15], *V/V*_0_ had a good change law with *L/d*; *d* is the equivalent diameter, which is the length of square hood side (*a*). It shows that the center-line velocity change regime can be obtained by dimensionless method, and it can eliminate the influence of hood face velocity and the hood size on the velocity change regime outside the exhaust hood; however the relationship between *V/V*_0_ with *L/d* was different for different type hoods.

As shown in Figure 4, when the distance was 0.1 times the length of the hood side, the velocity decreased to 90% of the hood face velocity. When the distance was twice the length of the hood side, the velocity decreased to 0.04 m/s, after which it remained approximately stable. Therefore, the study of the center-line velocity change regime of a parallel-flow square exhaust hood in a distance of twice the length of the square hood side from the hood face can satisfy engineering design needs and effectively control dust and toxic pollutants as long as the flow regime assumptions are satisfied by the pollutants.

When the velocity decreased to 10% of the hood face velocity, *L/d* (*d* is equal to *a* in a square hood) of a parallel-flow exhaust hood was 1.2, but *L/d* of a desktop slot exhaust hood without flange was about 1.5 [15], which shows that the control distance of the hood rest on the table is larger than that of the open hood, because the table can be considered a flange. Therefore, if a parallel-flow exhaust has a flange, it may also enhance the exhaust effect. When the velocity decreased to 10% of the hood face velocity, *L/d* of a parallel-flow supply hood was 24.6 [14], which is 20.5 times *L/d* of the parallel-flow exhaust hood. 

This study was performed to determine the characteristic changes of the center-line velocity of air in a parallel-flow square exhaust hood, and the research results can be used to guide exhaust hood controls of dust and toxic pollutants. To eliminate simulation and measurement errors, the average of the simulation results for different hood face velocities (0.5 m/s, 0.6 m/s, 0.7 m/s, and 0.8 m/s) were used. Thus, the average *L/a* and *V/V*_0_ values of different hood face velocities were used as the *x*-axis and *y*-axis, respectively, and was analyzed in Excel (Figure 5). 

After calculating the index, logarithm, linear, and polynomial trend lines, the reliability (*R*^2^ value) of the polynomial trend lines was 0.9986, therefore the polynomial trend line was found to most closely fit the data. Center-line velocity change regime in a parallel-flow square exhaust hood can be described by Equation (1).
*y* = −0.2778*x*^3^ + 1.246*x*^2^ − 1.8773*x* + 1.0176(1)
where *y* is *V/V*_0_ and x is *L/a*.

Both the parallel-flow exhaust hood studied in this paper and the desk-top slot hood studied in Wu et al. [15] are exhaust hoods, but their center-line velocity change regimes are different because of the different hood type. Both the parallel-flow exhaust hood studied in this paper and the parallel-flow supply hood studied in Chen et al. [14] were uniform flow, but their center-line velocity change regimes were quite different. 

### 3.3. Experimental Verification

In order to verify the accuracy of the regression Equation (1), and to investigate whether the influence of exhaust hood size on the center-line velocity change regime in a parallel-flow exhaust hood can be eliminated as the velocity change regime proposed via dimensionless for a parallel-flow square supply hood [14] and a desktop slot exhaust hood without flange [15], a 0.7 m (long) and 0.7 m (wide) parallel-flow exhaust hood commonly used in practice was taken as an example for the experimental study, which was different from the hood size used in the simulation. 

It is necessary to detect the velocity at multiple locations on the hood face in order to verify whether the air flow of the exhaust hood used in the experiment was uniform, because the numerical simulation assumes that the air flow in the hood has a parallel-flow. Sixteen velocity detection points were evenly arranged on the hood face. The velocities were detected by a 24-channel anemometer (Model 6242, Velocity sensor: 0965-03, KANOMAX Co., Takeda, Osaka, Japan, www.kanomax.co.jp). At each detection point, one velocity reading per second was taken and the average of 100 velocity readings was calculated for that point. The schematic diagram of velocity detection at the hood face of the experimental setup is shown in Figure 6.

The test results of velocity (*V_test_*) at the hood face are list in Table 2.

According to the test results of velocity in Table 2, the average velocity (*V_avg_*) of the test results was calculated. The velocity deviation was calculated by the formula of (*V_test_* - *V_avg_*)/*V_avg_*, and the results are shown in Table 3.

When the deviation between the max and min velocity and the average velocity is less than 50% [19,20], the air flow can be regarded as having a parallel-flow. It can be seen from Table 3 that the deviation between max velocity and average velocity was 14%, and the deviation between min velocity and average velocity was −13%. Therefore, it can be considered that the air flow at the face section of the experimental exhaust hood used in this study had a parallel-flow.

Nineteen center-line velocity detection points were set at regular intervals of 5 cm along the center line of the hood, starting from the hood face, until the detection result was basically the same or close to the detection sensitivity. The schematic diagram of center-line velocity detection setup is shown in Figure 7.

A comparison of detection and calculated results are shown in Figure 8.

As can be seen from Figure 8, the relative deviation between experimental data and calculated data ranges from −19.46% to 16.81% and the average relative deviation was 12.70%. The deviation range of velocity at the hood face was −13% to 14% in a parallel-flow exhaust hood used in the experiment of this study, which is basically the same as the average relative deviation between the experimental data and calculated data. Therefore, the experimental data is basically consistent with the calculated data, which shows that the regression equation method is reliable. 

It was assumed that the flow velocity at the hood face was uniform in this study, but there would be a separation of the flow that occurred when entering the exhaust [21,22,23], and a certain degree of non-uniformity for the velocity of a parallel-flow square exhaust hood in practice, such as the deviation range of velocity at the hood being 13% to 14% in a parallel-flow exhaust hood used in the experiment of this study. This may be the reason that the numerical simulation results are not completely consistent with the experimental results. Another reason is that there was a support at the bottom of the exhaust hood used in the experiment, which can be regarded as a flange and has a certain influence on the air flow. Therefore, the influence of the flange and the separation of the flow that occurs when entering the exhaust can be investigated in future studies.

The regression Equation (1) of numerical simulation, and a reasonably acceptable agreement with the experimental data, shows that the velocity and distance for a parallel-flow square exhaust hood can be expressed by Equation (1), which can eliminate the influence of the hood face velocity and the hood size on the center-line velocity change regime. Therefore, the center-line velocity change regime proposed in this paper can provide the technical basis for the application of a parallel-flow square exhaust hood in practice, so as to obtain the best control effect of dust and toxic pollutants.

## 4. Discussion

When the velocity decreased to 10% of the hood face velocity, *L/d* of a parallel-flow supply hood [14] was 20.5 times that of the parallel-flow exhaust hood. It shows that the control distance of the supply hood is far greater than that of the exhaust hood. Therefore, the control distance of contaminant in a parallel-flow push–pull ventilation system is mainly extended by the supply air as reported in previous literature [2,3,4,5,6]. Huang et al. [7] and Betta et al. [8] researched the aerodynamic characteristics and design guidelines of jet push–pull ventilation systems, and Cao et al. [9] analyzed the distribution of velocity and SF_6_ (sulfur hexafluoride) concentration for uniform flow and jet push–pull ventilation system. However, it was not clear whether the center-line velocity can be predicted by the dimensionless method for the uniform air flow push–pull ventilation system, and it should be researched in future studies.

The center-line velocity change regime in a parallel-flow square exhaust hood has been studied in this paper, but the air flow distribution outside the center-line of the hood should also be understood and studied in the future, especially when the contaminant source is large [13].

The results of this study can be used to determine whether a certain position satisfies the requirement for controlling velocity of a toxicant to provide a basis for the location of the source of the toxicant. This study is important for the practical application of parallel-flow square exhaust hoods and can provide a theoretical basis for the study of indoor ventilation for controlling toxicants in the field of occupational hygiene.

## 5. Conclusions

The conclusions of this study are as follows:The center-line velocity change regime in a parallel-flow square exhaust hood can be described by Equation (1), which is *y* = −0.2778*x*^3^ + 1.246*x*^2^ − 1.8773*x* + 1.0176, where *y* is *V/V*_0_, and *x* is *L/a*.The dimensionless center-line velocity (*V/V*_0_) has a good change law with the characteristic length of exhaust hood *(L/a**)* in a parallel-flow square exhaust hood, which can eliminate the influence of hood face velocity and the hood size on the center-line velocity change regime in a parallel-flow exhaust hood. It also shows that the dimensionless method can be used to study the variation of velocity in a parallel-flow square exhaust hood in order to get the general law of center-line velocity change.The experimental data are in good agreement with the numerical simulation results, showing the regression equation method is reliable, but the numerical simulation results are not completely consistent with the experimental results, and the reason why the two results are not completely consistent may be that there will be a separation of the flow that occurs when entering the exhaust and a certain degree of non-uniformity for the velocity of a parallel-flow square exhaust hood in practice, but it was ignored in this study.

## Figures and Tables

**Figure 1 ijerph-17-04485-f001:**
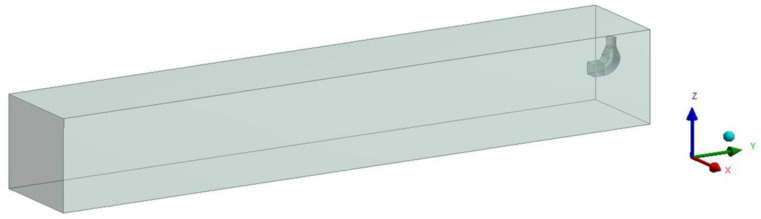
Geometric model and calculation field for the square exhaust hood.

**Figure 3 ijerph-17-04485-f003:**
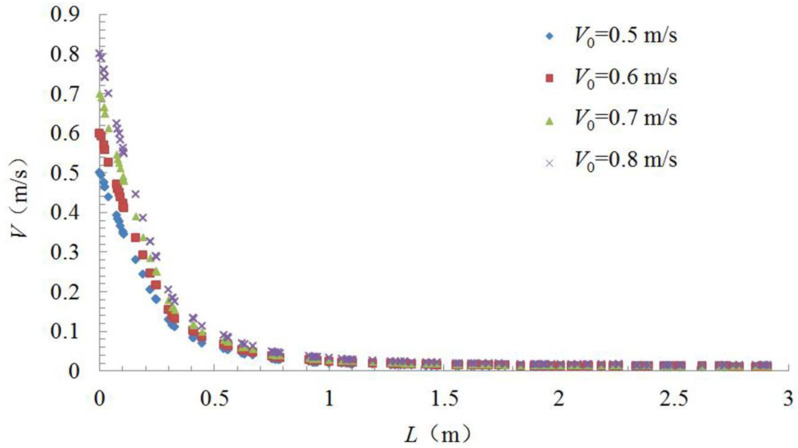
Change in hood axis velocity (*V*) with distance from hood face (*L*) for various hood face velocities (*V*_0_).

**Figure 4 ijerph-17-04485-f004:**
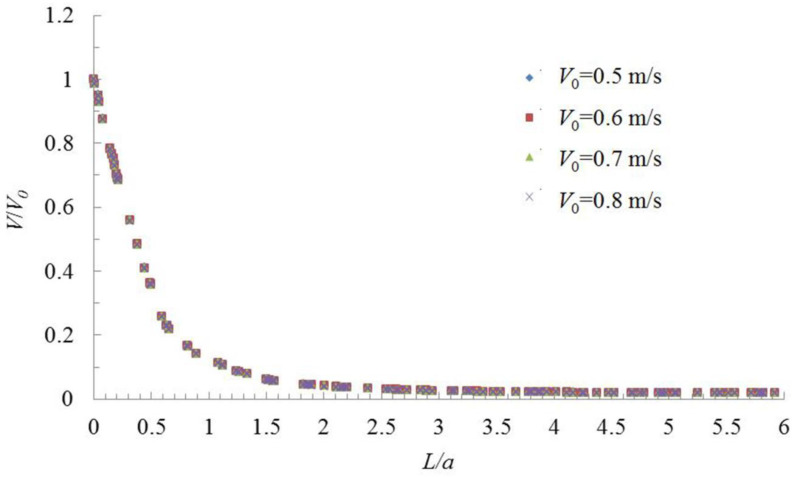
Dimensionless change in hood axis velocity (*V/V*_0_) with dimensionless distance from hood face (*L/a*) for various hood face velocities (*V*_0_).

**Figure 5 ijerph-17-04485-f005:**
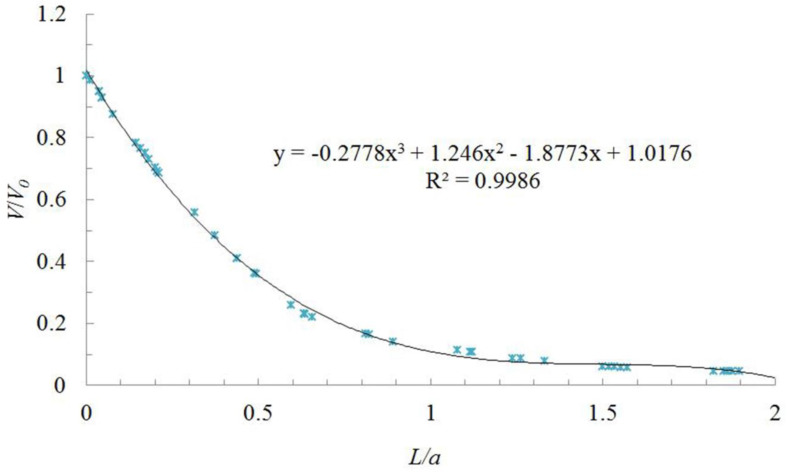
Dimensionless change in hood axis velocity (*V/V*_0_) for different dimensionless distances from hood face (*L/a*).

**Figure 6 ijerph-17-04485-f006:**
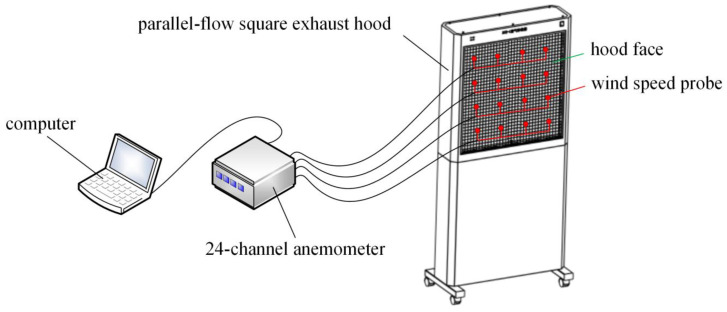
Schematic diagram of velocity detection at the hood face of experimental setup.

**Figure 7 ijerph-17-04485-f007:**
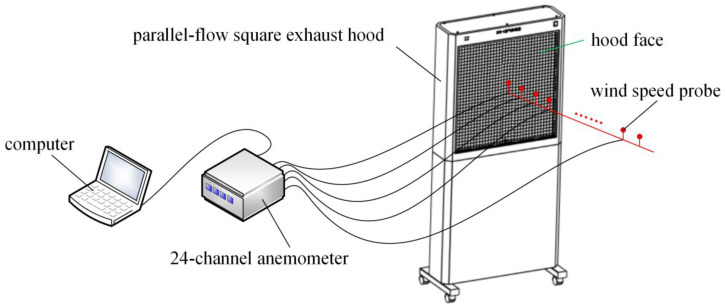
The schematic diagram of center-line velocity detection setup.

**Figure 8 ijerph-17-04485-f008:**
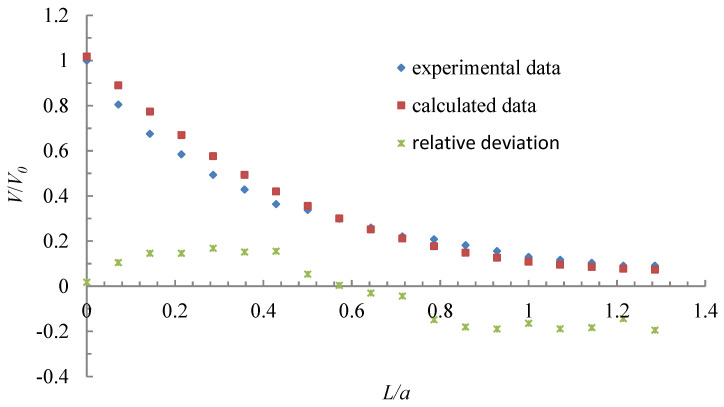
Comparison of experimental and calculated results.

**Table 1 ijerph-17-04485-t001:** Boundary conditions and solve define used in this study.

Boundary Conditions	Parameter Setting
Inlet boundary type	velocity inlet
Velocity inlet (m/s)	−0.5, −0.6, −0.7, and −0.8
Hydraulic diameter of inlet (m)	0.5
Material	Air
Air viscosity (kg/(m·s))	1.7894 × 10^−5^
Turbulence intensity of inlet (%)	4.85, 4.74, 4.65, and 4.58
Outlet boundary type	Outflow
Hydraulic diameter of outlet (m)	0.3
Solver	Segregated
Viscous model	k-epsilon
Energy equation	Off
Pressure–velocity coupling	Simple
Momentum	Second order upwind
Convergence criterion	10^−6^
Iterations to store and plot	1000

**Table 2 ijerph-17-04485-t002:** Test results of the hood face velocity.

Velocity (m/s)		Horizontal Detection Position			
		1	2	3	4
**Vertical detection position**	1	0.68	0.64	0.65	0.62
	2	0.8	0.74	0.75	0.78
	3	0.81	0.76	0.76	0.78
	4	0.64	0.63	0.65	0.67

**Table 3 ijerph-17-04485-t003:** Averages and deviations of the velocity.

	Average Velocity	Max Velocity	Min Velocity
**Velocity (m/s)**	0.71	0.81	0.62
**Velocity deviation**	–	14%	−13%

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
