# Peer review of "Center-Line Velocity Change Regime in a Parallel-Flow Square Exhaust Hood"

_ijerph, 2020, doi:10.3390/ijerph17124485_

Round 1

Reviewer 1 Report

Thank you for making all the suggested changes. The paper is in much better shape now. There are still some minor edits I would like to suggest.

  1. Line 29: change "improved" to "improves"
  2. Line 31: First word of this line "The" can be omitted
  3. Line 54: A short explanation can be made on how exactly can the control of pollutants be improved, along with any accompanying references.
  4. Line 56: A reference should be added after "...occupational health of workers".
  5. All the figures should be of higher image resolution.
  6. Lines 104 to 109 can be eliminated since the same explanation is repeated in lines 128 to 131 below.
  7. Line 113: "....and 0.8 m/s)" should end with a full stop instead of a semicolon. The next sentence should begin with a capital letter.
  8. Line 120: replace the term "wind speed" with "face velocity" to make the terminology consistent with that used elsewhere, for example line 96.
  9. Line 147: It may be necessary to say "as long as the flow regime assumptions are satisfied by the pollutants" since larger dust particles may have significant inertia and viscous resistance associated with may need more than 0.04 m/s flow velocity to induce physical movement of the particles.
  10. Line 152: Replace the beginning two words "with the" with "has a"
  11. Line 155. insert the word "performed" after "This study was"; insert "of air" after "the center-line velocity"
  12. Line 156: De-capitalize the letter "E" in "Exhaust" and insert "of" after "controls".
  13. Line 175: Please rewrite the sentence as "...of the regression equation (1) and to investigate whether the ..."
  14. Line 178: Please rewrite as "a 0.7-m-long and 0.7-m-wide ....was taken" instead of writing "take a .."
  15. Line 181: Insert "multiple locations" after "..velocity at"
  16. Line 183: "16 velocity detection points were...", not "are"
  17. Line 184: after the name of the instrument at the end of the line, include in parentheses the name of the manufacturer and location so that other researchers can try to replicate your experiments buy buying the same exact instruments. Also include any registry (R) sign or a trademark (TM) sign associated with the make/model along with any references to the respective website or brochure.
  18. Lines 185 and 186: rewrite this sentence as "At each detection point, one velocity reading per second was taken and the average of 100 velocity readings were calculated for that point."
  19. Lines 193 and 194: after "...Table 2," replace the phrase with "...in Table 2, the average velocity...was calculated...".
  20. Table 3: replace "wind speed" with "velocity" to make it consistent with Table 2 terminology. Also, get rid of "the" before "velocity deviation" in the first column
  21. Line 199: Insert a reference after "...parallel-flow". According to which standard is this statement made?
  22. Line 203: Insert "center-line" after "The" at the very beginning of this line. Also replace "..set at a distance of " with "...set at regular intervals of 5 cm", also specifying the total number of center-line velocity measurement points.
  23. Line 218: Replace "wind speed" with "flow velocity" to make the terminology consistent. Also replace "the" at the end of the line with "a"
  24. Line 226: Replace "should be studied later" with "can be investigated in future studies."
  25. Line 227: Insert the equation number after "The regression equation". Also, replace "experiments" with "a reasonably acceptable agreement with the experimental data"
  26. Line 243: Replace "...the further" with "future studies".
  27. Line 256: Replace "V/Vo" quotes with parentheses: (V/Vo) for consistency. Similarly, for L/a in line 257

Author Response

I wish to resubmit the manuscript, titled “Center-line velocity change regime in a parallel-flow square exhaust hood.” The manuscript ID is ijerph-844270.

Thank you very much for your professional, patient and meticulous comments. Not only this paper is well modified, but also my writing ability has been greatly improved. Thank you again.

I have revised your opinions point by point. Please see the attachment for details of modification.

Reviewer 2 Report

After the last round of revision, I think the quality of this article has improved.

It is only recommended to improve the clarity of the pictures.

Author Response

(The authors gave the same response as above.)

Reviewer 3 Report

These authors have justified the initial concern in terms of the rationale.  The center-line velocity decreases rapidly with increasing distance from the hood face, because the velocity decreases to approximately zero at a distance of 1.5 m from the hood face for all hood face velocities. When the hood face wind speed is different, the variation law of the center-line velocity is different, which is consistent with the research conclusion in two of their previous reports. Therefore, they are somewhat justified in using the same dimensionless method to study the present parameters as well as further explore deviations/variations.

The authors have done an excellent job of clarifying aspects of the manuscript that are easily followed by track changes and subsequently is acceptable for publication in IJERPH. 

Author Response

I wish to resubmit the manuscript, titled “Center-line velocity change regime in a parallel-flow square exhaust hood.” The manuscript ID is ijerph-844270.

Thank you very much for your professional, patient and meticulous comments. Not only this paper is well modified, but also my writing ability has been greatly improved. Thank you again.

I have revised your opinions point by point. Please see the attachment for details of modification.

This manuscript is a resubmission of an earlier submission. The following is a list of the peer review reports and author responses from that submission.

Round 1

Reviewer 1 Report

This work uses the dimensionless method to study the center-line velocity change regime in a parallel-flow square exhaust hood based on the simulation and experimental data. The results show that V/V0 has a good change law with L/a in a parallel-flow square exhaust hood, and the experimental data are in agreement with the numerical simulation results, indicating that the regression equation method is reliable. The conclusions are supported by the data. It can be accepted after addressing the following issues: 1. The current status of parallel-flow square exhaust hood is not well reviewed. 2. The results should be compared with the literature. 3. The language should be carefully polished, they are lots of typo and grammatical errors.

Reviewer 2 Report

Major comments:

  1. Purpose is unclear. The main purpose of doing the simulations and backing it up with experimental data should be explained more clearly i.e., what is the main motivation of carrying out this study and what are the benefits of doing so.
  2. A figure needs to be added (at least in schematic form) describing the experimental setup that was used to validate the simulation findings.
  3. Figure of the simulation geometry should be labelled showing L, a, V and Vo clearly. 
  4. Further description of boundary conditions used in the computational domain walls is necessary (zero velocity/zero pressure/periodic boundary etc.). Also, rationale of using 25m X 4m X 4m should be given for the computational domain.
  5. Major changes in written English is necessary to convey the message more clearly
  6. Justification might be necessary of why the simulation does not take into account a table at the bottom of the computational domain (with a wall boundary condition of v=0). If such an exhaust system is to be brought to real-world application, the analysis presented in this paper would be questionable, unless this paper explicitly states that the exhaust system described in this paper is not to be used over a table, and only as a free and stand-alone unit hanging in a large room. Otherwise, the analysis should incorporate a zero-velocity table surface close to the bottom edge of the exhaust (bottom side of the rectangular exhaust face).
  7. Justification is necessary for using 0.7 m X 0.7 m experimental exhaust hood (why this size and why not any other size?)
  8. I disagree that Figure 6 shows good agreement between modeled and experimental values. There is clearly a systematic bias. The paper should at least acknowledge this bias and say that the simulated values are within x% of the experimentally observed ones

Minor/technical comments:

  1. Numerous typos
    • Table 1 line 8 from the top should be 'hydraulic diameter of outlet', not 'inlet'
    • Table 2 header on the right hand side should not have '(m/s)'. Also last line of Table 2 should say 'min velocity (m/s)', not 'max velocity (m/s)'
    • Line 179: "...regime proposed in this paper can provided ...." should be "...can provide..."
    • Line 187: "...it can eliminated.." should be "..it can eliminate..."
  2. Numerous grammar errors
  3. Need for further explanations of technical terms in one or two lines
  4. Figure 2 should be zoomed in and caption should say 'top view'
  5. Table 2 should be split into two parts: one for velocities in various horizontal and vertical positions, and one for averages and deviations

Detailed comments by line number:

  • Lines 13 and 14: Rewrite the sentence to make the problem more clear to the reader
  • Line 17: V, Vo, L and a are not described in the abstract. These either need to be defined, or alternatively use terms such as 'non-dimensional forms of face velocity and characteristic hood length' (or anything better)
  • Line 18: Wrong grammar use - 'which can eliminate', not 'eliminated'
  • Line 192: Define what you mean by 'desktop effect' in one or two sentences.
  • Line 195: Perhaps use 'contaminant' in the last word of the line instead of 'poison'
  • Line 200: instead of using the word 'obtained', it should rather be 'predicted'
  • Rewrite the sentence in line 203 to 205. It does not make clear sense.
  • Line 218: Eliminate the influence of hood face velocity and hood size on what? Rewrite the sentence to describe in detail what are you after, what are you trying to reduce and why.

Reviewer 3 Report

When reviewing scientific papers for publication, I usually start with a general overview in terms of a structure, abstract, literature review, methodology, findings of the research, discussion, conclusions, as well as limitations of the study.

In the assessment of the paper submitted for the review, I specifically focussed on the discussed issues, applied research methods and the scope of analysis of research results, as well as substantive content of the article and its structure.

The manuscript is interesting and original. It deals with an interesting topic which is currently of interest among researchers.

To improve the quality of the work I would recommend:

  1. Figure 2 to Figure 6 May have better performance.
  2. Line174 lacks definitions and standards for the reliability of regression analysis results.
  3. Supplementary research limitations and planning and recommendations for follow-up studies.
  4. Conclusion need to be more specific description and summary.

Reviewer 4 Report

All in all, centerline velocity and sidewall deposition are not well characterized and should be accounted for in the model put forth by these authors.  Further, the authors use the midstream velocity which is an adequate simplifying assumption for design purposes, however, there are known sidewall deposition and retention dynamics that occur in which midstream velocity cannot account for. The authors fail to account for these and should explain why their estimates are devoid of these additional dynamics for in-duct transport.